# Swimming Activity Alleviates the Symptoms of Attention: Deficit Hyperactivity Disorder (ADHD) a Case Report

**DOI:** 10.3390/healthcare11141999

**Published:** 2023-07-11

**Authors:** Smaragda Skalidou, Andreas Anestis, Nicoletta Bakolas, Georgia Tsoulfa, Konstantinos Papadimitriou

**Affiliations:** 1Laboratory of Hygiene, Social-Preventive Medicine and Medical Statistics, Faculty of Medicine, Aristotle University of Thessaloniki, 54124 Thessaloniki, Greece; sskalidou@gmail.com (S.S.); aanestis@act.edu (A.A.); 2Division of Technology and Sciences, The American College of Thessaloniki, 55535 Thessaloniki, Greece; gtsoulfa@anatolia.edu.gr; 3Independent Researcher, Bristol BS15LN, UK; nbakola@hotmail.com; 4Faculty of Health and Rehabilitation Sciences, University of East London, Metropolitan College of Thessaloniki, 54624 Thessaloniki, Greece

**Keywords:** physical activity, Beck Depression Inventory, Barratt Impulsiveness Scale, hospital anxiety and depression scale

## Abstract

Attention deficit hyperactivity disorder (ADHD) is a neurobehavioral disorder characterized by inattention, hyperactivity, and impulsivity. Sport and physical activity have been shown to play a major role in the development of cognition, memory, selective attention, and motor reaction time, especially among adolescents with ADHD. In this context, the objective of this study was to investigate the effects of a swimming exercise program on the symptoms of ADHD in an adult with a diagnosis since childhood. The training intervention was performed for eight weeks, and the results demonstrated that the swimming–learning program significantly alleviated the symptoms of inattention and hyperactivity, as measured by the psychometric indices used in the study. Further studies are needed to establish and understand the association between physical activities and improved mental performance in adults with ADHD.

## 1. Introduction

Attention deficit hyperactivity disorder (ADHD) is a neurobehavioral disorder that typically emerges during childhood, with prevalence estimates ranging from 2 to 18% globally and three times higher in males than females [1]. This wide range in prevalence has been explained based on cultural variations [2] and methodological limitations, such as the measurement of aggression, which may be expressed among girls in a different way than among boys [3]. Moreover, it affects approximately 3 to 10% of the school-aged population, making it the most common neurodevelopmental disorder of childhood and often continues into adulthood.

ADHD is characterized by three primary behavioral symptoms: (i) inattention, (ii) hyperactivity, and (iii) impulsivity [4]. This disorder significantly impacts individuals due to deficits in cognitive (e.g., difficulty in remembering, learning, and concentrating on new things) and motor functions (e.g., difficulty in simple or complex activities such as writing or competition in a sport, respectively), which can hinder their personal, social, and work lives. Abilities such as sustaining attention, reducing distractibility, and increasing response control are particularly important in these domains [5].

Current treatments for ADHD include pharmacological, psychotherapeutic, and psychoeducational interventions [6]. Nazarova et al. [7] found that almost in 20% of the recorded studies, there was a subscription of drugs such as central nervous system stimulants (e.g., methylphenidate hydrochloride, lisdexamfetamine dimesylate, amphetamine sulfate, mixed amphetamine salts, a combination of dexmethylphenidate hydrochloride and serdexmethylphenidate chloride), selective noradrenaline reuptake inhibitors (atomoxetine, viloxazine), and alpha2 adrenergic receptor agonists (guanfacine hydrochloride, clonidine hydrochloride). However, due to the ineffectiveness of medication in some cases, the presence of unpleasant side effects, or personal preference and high cost, alternative treatment approaches are constantly being explored [6,8].

Physical activity has been extensively studied in children with ADHD and has shown positive effects on cognitive performance, ADHD symptoms, motor abilities, mental health, cognitive functioning, and motor coordination. The mechanisms underlying the beneficial effects of exercise are attributed to increases in arousal and catecholamine levels. These mechanisms are associated with the functioning of the prefrontal cortex, which could explain why executive functions appear to benefit more from exercise compared to other cognitive functions. Additionally, there is evidence suggesting that exercise leads to increases in cerebral blood flow (CBF), potentially affecting prefrontal brain regions or overall CBF [9,10,11,12,13].

However, there is a limited body of research investigating the impact of physical activity, specifically in adults with ADHD. Nevertheless, existing studies suggest that even a single session of exercise can have positive effects on adult patients with ADHD [14].

Swimming is an aerobic-based activity that allows people to develop good coordination, as it requires synchronized movements with their arms and legs while providing psychological benefits [15]. Both children and adults seem to prefer participating in swimming training because it provides the opportunity to think creatively and to learn without recurring exhausting exercises [16]. Therefore, perhaps, children with ADHD find swimming sport interesting and preferable compared to other disciplines. Also, such movements are speculated to cause alterations in brain functioning [15,17]; specifically impacting the brain areas that underlie symptomatology, namely the prefrontal cortex and amygdala [17].

Hattabi et al. [18] and Silva et al. [19] demonstrated the beneficial effects of swimming training, compared to control groups, on ADHD symptoms in pre-adolescent cohorts. Specifically, in a 12-week and 8-week swimming training intervention, the participants showed improvements in cognitive function, behavior, academic performance, cognitive flexibility, and selective attention. Additionally, there was a reduction in depression parameters and stress levels. Furthermore, improvements were observed in motor coordination and physical fitness, including coordination of lower limb laterality, flexibility, and abdominal resistance.

Despite the positive conclusions regarding the effectiveness of swimming in reducing ADHD symptoms, the literature still lacks studies conducted on adult cohorts. Therefore, within this theoretical framework, the aim of the present case study was to examine the influence of an aerobic type of physical activity on ADHD symptoms. More specifically, an intensive swimming training program was designed with the goal of potentially alleviating ADHD symptoms and enabling a 31-year-old female to effectively handle her workplace responsibilities.

## 2. Case Presentation

The subject of this case study was a 31-year-old woman (body height: 157 cm, body weight: 65 kg, and BMI: 26.4 kg/cm^2^) experiencing difficulties in concentrating and maintaining attention on tasks of her job and daily routine activities such as house cleaning, cooking, etc. More specifically, she had forgotten to close the oven in the kitchen while she was trying to solve a problem with the placement of the dishes in the cupboard. This situation made her nervous, and as a result, she felt depressed and unsuitable to deal with this.

The first appearance of ADHD was observed when she was eight years old, and she was diagnosed with it. A recent assessment, which was conducted before intervention, via Conner’s continuous performance test (CCPT) for adults [20], indicated the presence of a combined type of ADHD (T-score ≥ 60). The patient presented four symptoms from the inattention list and more than four from the hyperactivity and impulsivity list of the Diagnostic and Statistical Manual of Mental Disorders (DSM-5) test [4]. She had not been diagnosed with any other medical or mental disorder which could better explain her symptoms.

During the study, she was working as a nurse; hence, the symptoms manifested more frequently at the workplace and in intense, stressful, and emotional situations. On the contrary, no such symptoms were observed when the patient engaged in activities she enjoyed, such as long-distance running.

The patient followed an intense swimming training program for eight consecutive weeks, from 10 October to 9 December 2022, with three swimming and two dryland training sessions per week. It was held on Monday, Wednesday, and Friday from 9:00 a.m. to 12:00 p.m. to avoid the circadian rhythm problem [16]. Each session lasted 90 min and was divided into three stages: a 15 min warm-up, 70 min of aquatic exercises, and a 5 min cool-down. The intensity of training was assessed using the Ratio of Perceived Exertion (RPE) scale from 0 to 10 [16]. It was calculated for each training session to be near 7 and 8 on the RPE scale. The program was implemented at a public 25 m, 11 lanes, open swimming pool in Thessaloniki, Greece. The pool’s temperature was maintained at 29 ± 1 °C, and decontamination occurred at the end of each week.

As the individual did not know how to swim before the intervention, the instruction started with simple tasks such as head immersion and exhalation under the water, flutter kicks in the front and supine position, etc. Then, the different hand strokes and swimming styles were taught and practiced. Also, the burden of the activity was increased by combining different swimming strokes and increasing its total training volume.

In addition, each new technique was introduced once the previous one had become a subconscious movement through repetition. Staying focused, memorizing the details of the technique, and practicing helped the patient achieve the goals of each training session. Moreover, the two dryland sessions included swimming technique drills, enhancing her ability to understand the drills in swimming sessions. The volume of each exercise repetition was 25 m, while the sets per exercise varied from four to six. The content of the drills is presented in Table A1 (Appendix B).

While participating in this sport, she discovered that her ADHD symptoms were more manageable, and she attributed this to her rigorous exercise routine. However, after the eight-week intervention, she was forced to discontinue the sport due to an injury. As a result, she experienced an increase in ADHD symptoms, which significantly impacted the quality of her work.

The patient underwent evaluation using the following validated psychometric scales after obtaining informed consent: (i) the Beck Depression Inventory (BDI-II), (ii) the Barratt Impulsiveness Scale (BIS-11) [21], and (iii) the Hospital Anxiety and Depression Scale (HADS-14) [22]. The measurements were conducted prior to the intervention, in the middle (fourth week), and at the end of it (eighth week). Unfortunately, BIS-11 was not answered by the patient because she was not in the mental condition to answer all the questions. Therefore, we avoided putting pressure on her. BDI-II [23] (Appendix A) is a self-report questionnaire consisting of 21 sets of statements designed to discriminate subtypes of depression and differentiates depression from anxiety (e.g., guilt, low self-worth, irritability, and suicidal ideation). Each set is ranked in terms of severity and scored from 0 to 3. Scores of 0–9 fall into the normal range, 10–18 indicate the presence of mild to moderate depression, 19–29 indicate moderate to severe depression, and 30–63 indicate severe depression. The BDI-II demonstrates high internal consistency, with alpha coefficients of 0.86 and 0.81 for psychiatric and non-psychiatric populations, respectively. The test offers a reliable and valid index to discriminate between depressed and non-depressed subjects portraying depressive symptoms and attitudes which can be used effectively to document changes brought about in therapy. Also, the BDI-II is a cost-effective questionnaire for measuring the severity of depression, with broad applicability for research and clinical practice [24]. The BIS [25,26] (Appendix A) is a self-report test used to evaluate the personality/behavioral construct of impulsiveness. It is a questionnaire with 30 items that are scored on a 4-point scale (Rarely/Never = 1, Occasionally = 2, Often = 3, Almost Always/Always = 4). Scores of 4 indicate the most impulsive response. The higher the summed score for all items, the higher the level of impulsiveness. Items include statements such as “I do things without thinking or I concentrate easily”. Its psychometric properties have been determined in both clinical and non-clinical subjects [27], showing its usefulness in psychiatric, scientific, and clinical conditions, evaluating the impulsivity construct in different ethnicities [28].

The HADS-14 [20] (Appendix A) is also a self-assessment questionnaire that detects states of anxiety and depression. The questionnaire has seven items each for depression and anxiety subscales, respectively. Scoring for each item ranges from zero to three, with three denoting the highest anxiety or depression level. A total subscale score of more than 8 points out of a possible 21 denotes considerable symptoms of anxiety or depression. HADS-14 shows significant diagnostic validity and reliability and is suggested to be used in clinical monitoring of the psychiatric and psychological status of different kinds of patients [29].

The patient’s psychometric tests showed a significant percentage reduction. Specifically, BDI-II scores, prior to the intervention until immediately after it, dropped from mild depression to normal, respectively, showing a 100% reduction. Also, the BIS-11 score decreased by almost 50% compared to the beginning of the intervention, indicating an individual that is extremely over-controlled. Concluding, the score in HADS-14 revealed that the patient, from a situation highly likely for a mental disorder, reached a lack of it at the end of the intervention with a reduction near 33% (Table 1).

## 3. Discussion

The aim of the present case study was to investigate the effect of swimming, a closed motor skills sport, on the symptoms of ADHD in an adult who had been diagnosed with the condition since childhood and was not taking any medication. It is generally well-established that closed motor skills are beneficial for hyperactive/impulsive symptoms, while open motor skills are beneficial for attention problems [30].

According to our results, it was found that the psychological condition of the subject improved during and after the exercise program. In fact, her scores on all three psychometric scales (BDI-II, BIS-11, HADS-14) used depleted to levels indicative of someone with no ADHD, demonstrating a decrement from 33 to 100%. Ahmadi et al. [31] showed that female students who participated in swimming sports for four weeks showed a drop in their mean depression scores (BDI-II) from 10.5 to 6.5, demonstrating the beneficial role of exercise. Similarly, in our study, BDI-II scores dropped from 12 to 0 in the first four weeks of the intervention and remained low until the end of it (eighth week).

Different results were observed in the BIS-11 scores compared to the literature. Specifically, in our case report, the participant significantly reduced her scores at the end of the exercise intervention from 57 to 27. Lee et al. [32] showed that the physical exercise group increased their BIS-11 score from 49.23 to 51.18, compared with a self-management and liberal arts group, concluding that a self-management course ameliorates this index more effectively in groups of adults. The given explanation for this contradiction focused on the intensity of the exercise. Lee et al. [32] suggested that the intensity of activities, which were resistance and cardio exercises, were not high enough to show a significant difference. In contrast, in our study, certain parts of the training were conducted at a high intensity. As a result, this difference is reflected in the lower BIS-11 score.

Last but not least, the HADS-14 scores decreased during and after the intervention program, indicating a lower depression and anxiety profile. Similarly, de Oliveira et al. [33] demonstrated a correlation between high levels of physical activity and a lack of symptoms of anxiety and depression in an elderly sample. Specifically, in our study, the score decreased from 21 to 7, while in the elderly sample, the HADS-A and D scores decreased from 5.23 and 5.81 to 3.78 and 3.01, respectively. However, the HADS scores in de Oliveira’s study were already low at the beginning of the intervention. Also, the sample consisted of older adults, which suggests that the influence of exercise is likely greater when compared to our younger participants.

Regarding the participant’s personality, her dedication to focusing on the required tasks and her commitment to practicing diligently to achieve her goal of learning to swim and training as a long-distance swimmer contributed to her successful performance in her professional responsibilities as a nurse. Furthermore, as reported by the participant, engaging in physical activity helped alleviate her intense ADHD symptoms and deal with her professional responsibilities more effectively.

In particular, the patient approached swimming training not just as a hobby but as a form of therapy to help manage her ADHD symptoms. This decision was based on her previous experience with sports, where engaging in vigorous physical activities during childhood had proven beneficial in controlling her ADHD symptoms. Therefore, it is plausible that a closed motor skill activity like swimming can have positive effects on ADHD symptoms like hyperactivity/impulsiveness, confirming the literature [18]. Additionally, the patient experienced a reduction in attention problems, which is contradictive according to the literature, which suggests that patients benefitted from attention problems only in open motor skills [19].

Perhaps the repeatability of movements involved in closed motor skills, such as swimming, can be beneficial for ADHD patients. These skills provide precise movements that are repeatedly practiced, making them easier to learn and concentrate on specific stuff. Open motor skills, on the other hand, may be more challenging for patients due to the requirement of multiple movements, which can lead to confusion and distraction. Therefore, our finding presents another point for discussion and a perspective that warrants exploration.

In our case, the patient’s questionnaire scores indicated an improvement in her mental health. Furthermore, by observing her swimming ability and reported behavior, it was assumed that this swimming intervention had a positive influence on cognitive functioning, motor coordination, and physical fitness. However, it is challenging to generalize our results due to the necessity of larger cohorts in order to assess the impact of swimming activity on ADHD.

Considering the literature, it is indicated that physical activity has a positive impact on mental health, cognitive functioning, motor coordination, and physical fitness in individuals with ADHD [34] as well as in patients with other diseases [35]. Also, meta-analyses of two group control studies have further confirmed that physical activity interventions significantly improve inattentive symptoms [30] and can be used as a strategy to support adults [14] and children [30] in overcoming ADHD deficiencies. The present case report sheds light on an unexplored condition primarily studied in children. However, further studies need to be conducted to demonstrate the beneficial role of physical activity and determine whether aerobic is superior to non-aerobic types of exercise on ADHD symptoms [36]. According to Stein et al. [36], both aerobic and non-aerobic groups benefitted significantly, reducing self-reported depression in comparison to the controls. Also, it is noted that the non-aerobic was superior to the aerobic exercise for enhancing self-concept. Therefore, future intervention studies that incorporate a combination of aerobic and non-aerobic exercises will provide clearer insights into the precise exercise prescription for children and adults with ADHD.

## 4. Conclusions

An eight-week vigorous swimming intervention appears to alleviate symptoms of inattention and hyperactivity on all three psychometric scales, BDI-II, BIS-11, and HADS-14, improving the mental health of a 31-year-old female patient. However, based on the limitations of our study, it is concluded that further intervention studies are needed to establish and understand the association between physical activities and improved mental performance in larger adult cohorts with ADHD, utilizing more psychometric and performance measurements.

## 5. Study’s Limitations

In the present case, the intervention of an ADHD adult patient was not found in any other study. Thus, the evaluation of mental health and the observation of her behavior is novel. Despite this, there are some limitations in the study that probably affect the results.

We required additional measurements related to physical performance and cognitive function. The reason behind this decision was to minimize disturbances and potential anxiety that these tests could cause by using a reduced number of measurements;We did not use the BIS-11 questionnaire in the middle measurement of the intervention because she was not in the mental condition to answer all the questions. Therefore, we avoided putting pressure on her.

## Figures and Tables

**Table 1 healthcare-11-01999-t001:** BDI-II, BIS-11, and HADS-14 self-reporting scores prior to, during, and immediately after the intervention (intensive swimming exercise program).

Psychometric Test	Score		
	Prior Intervention (0 Week)	During Intervention (4th Week)	Immediately after Intervention (8th Week)
BDI-II	12	0	0
BIS-11	57	(Not performed)	27
HADS-14	21	7	7

BDI-II: Beck Depression Inventory; BIS-11: Barratt Impulsiveness Scale; HADS-14: Hospital Anxiety and Depression Scale.

## Data Availability

The authors confirm that the data supporting the findings of this study are available within the article.

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
