# Peer review of "Swimming Activity Alleviates the Symptoms of Attention: Deficit Hyperactivity Disorder (ADHD) a Case Report"

_healthcare, 2023, doi:10.3390/healthcare11141999_

Round 1

Reviewer 1 Report

Although the topic is interesting, the manuscript does not meet the requirements for publication, and therefore it is rejected. 

The manuscript does not provide sufficient background. It is not clear what theoretical framework supports the study.

Sometimes extrapolations are made without a theoretical background. For example, in line 41 of the introduction it is mentioned “Physical activity has been extensively studied in children with ADHD and has shown positive effects on cognitive performance, ADHD symptoms, motor abilities, mental health, cognitive functioning, and motor coordination”.

It is not clear from the methodology why the BIS-11 was not applied during the intervention.

In line 189 it is stated: "Considering the results of our case study, these align with the existing literature, which indicates that physical activity has a positive impact on mental health, cognitive functioning, motor coordination, and physical fitness in individuals with ADHD". Based on the data collection instruments used, it is not clear how the authors inferred improvements in cognitive functioning, motor coordination, and physical fitness.

The conclusions are too sparse and are not seriously reflected based on the limitations of the study developed.

The bibliographical references are sparse and could be more related to the article. References 8 and 11 are the same.

Author Response

Reviewer 1

Dear reviewers

We really appreciate the time spent by the judge of our manuscript. Your instructions helped us to improve the manuscript.

C: Comment

A: Answer

C: The manuscript does not provide sufficient background. It is not clear what theoretical framework supports the study.

A: We included the theoretical framework of our study. Specifically, we demonstrate the beneficial effects of swimming training in pre-adolescent cohorts and the lack of studies in adults (lines 78 – 93).

C: Sometimes extrapolations are made without a theoretical background. For example, in line 41 of the introduction it is mentioned “Physical activity has been extensively studied in children with ADHD and has shown positive effects on cognitive performance, ADHD symptoms, motor abilities, mental health, cognitive functioning, and motor coordination”.

A: We have included a theoretical framework, specifically, outlining the probable mechanisms of exercise and drawing conclusions about its effects on the adult cohort (lines 55 – 68).

C: It is not clear from the methodology why the BIS-11 was not applied during the intervention.

A: We clearly understand your concern. This is a limitation of our study which specifies her mental condition. As a result, she was not in the mood to answer all the questionnaires. We have included a limitation section in our study (lines 279 – 281). Also, we included the condition in the description of the questionnaires (lines 144 – 145).

C: In line 189 it is stated: "Considering the results of our case study, these align with the existing literature, which indicates that physical activity has a positive impact on mental health, cognitive functioning, motor coordination, and physical fitness in individuals with ADHD". Based on the data collection instruments used, it is not clear how the authors inferred improvements in cognitive functioning, motor coordination, and physical fitness.

A: You are absolutely right. We attempted to formulate a hypothesis by comparing her attitude, behavior, and performance in training from the beginning to the end of the program. However, we acknowledged this as a limitation of the study, explaining that conducting a more significant number of measurements could potentially cause stress for her. Therefore, we opted to avoid such measurements to ensure her well-being (lines 292 – 294)."

C: The conclusions are too sparse and are not seriously reflected based on the limitations of the study developed.

A: We modified the conclusion, portraying the results and limitations of our study (lines 280 – 287).

C: The bibliographical references are sparse and could be more related to the article. References 8 and 11 are the same.

A: We enriched the literature with related studies and corrected the references 8 and 11.

Reviewer 2 Report

PEER REVIEW PROCESS

Dear Authors,

Thank you for your lovely work. Here there are some suggestions to improve your manuscript.

TITLE

The title must underline that this is a case study. So even if you wrote "(…) of an adult patient", I suggest you indicate the form "a case study" directly in the title. 

INTRODUCTION

The introduction section is a little bit "thin" and essential. Therefore, I suggest you improve the introduction by trying to answer the following questions:

Why do people with ADHD prefer swimming instead of other disciplines?

What other domains (gym activities, running, cycling, horse riding etc.) were compared with swimming? (If they were, if not? Why?)

About pharmacological therapy: What are the most common drugs that are usually given to these people in combination with physical activity? (if there are any if not, declare them)-

CASE PRESENTATION AND CONCLUSION

These two sections are well-written and explained.

APPENDIX

I suggest you insert tables related to the questionnaires you administered during the experimental phase. In particular, underline the items and the scores at the study's beginning and end (T0, T1, etc.). In addition, please insert the percentage difference among measurements that is attractive to quantify the treatment results. Finally, I suggest you put it (% difference) in the text during the case discussion.

The quality of English is fine. Therefore, only minor editing is required.

However,  I am not an English native speaker. Thus, I suggest you ask a native speaker for a profound revision to ensure that the use of English is acceptable.

Author Response

Reviewer 2

Dear reviewers

We really appreciate the time spent by the judge of our manuscript. Your instructions helped us to improve the manuscript.

C: Comment

A: Answer

C: The title must underline that this is a case study. So even if you wrote "(…) of an adult patient", I suggest you indicate the form "a case study" directly in the title. 

A: Done. The title now is “Swimming Activity Alleviates the Symptoms of Attention – Deficit Hyperactivity Disorder (ADHD). A Case Study.”

C: The introduction section is a bit "thin" and essential. Therefore, I suggest you improve the introduction by trying to answer the following questions:

  • Why do people with ADHD prefer swimming instead of other disciplines?
  • What other domains (gym activities, running, cycling, horse riding, etc.) were compared with swimming? (If they were, if not? Why?)
  • About pharmacological therapy: What are the most common drugs that are usually given to these people in combination with physical activity? (if there are any if not, declare them)-

A: Thank you for your guidance. We enriched the introduction, answering your questions (lines 30 – 93).

C: I suggest you insert tables related to the questionnaires you administered during the experimental phase. In particular, underline the items and the scores at the study's beginning and end (T0, T1, etc.).

A: We included as supplementary files the three questionnaires that were used. Also, in the case presentation, we provide the table 2, depicting the scores during the intervention.

C: In addition, please insert the percentage difference among measurements that is attractive to quantify the treatment results.

A: We inserted the percentages in the text of the results, above the table 2 (lines 179 – 185).

C: Finally, I suggest you put it (% difference) in the text during the case discussion.

A: We included the percentages in the discussion (line 202).

Reviewer 3 Report

the paper is well structured designed.

the main issue here is being a study case of one person only, and with these results we cannot know if with more patients the results would be similar.

considering that this study case can be important it has enough quality for publication.

i would change the title to: Swimming Activity Promotes Relief for Symptoms of Atention – 2 Deficit Hyperactivity Disorder (ADHD): The Incidence of An 3 Adult Patient

the english quality in enough.

Author Response

Reviewer 3

Dear reviewers

We really appreciate the time spent by the judge of our manuscript. Your instructions helped us to improve the manuscript.

C: Comment

A: Answer

C: The main issue here is being a study case of one person only, and with these results we cannot know if with more patients the results would be similar.

A: We understand your concern. Therefore, we have stated carefully that our results are a first step for further studies of larger cohorts. Specifically, we have pointed this out in the discussion and conclusion (lines 261 – 263 & 281 – 287).

C: i would change the title to: Swimming Activity Promotes Relief for Symptoms of Atention – 2 Deficit Hyperactivity Disorder (ADHD): The Incidence of An Adult Patient

A: We really respect your suggestion for the improvement of our title. Another reviewer suggested a modification in our title too. We are confident that this change accurately describes the content of the study.

“Swimming Activity Alleviates the Symptoms of Attention–Deficit Hyperactivity Disorder (ADHD). A Case Study.”

Reviewer 4 Report

Thank you for the opportunity to review this paper. This study investigated the effects of an 8-week swimming exercise program on ADHD symptoms in an adult with a long-standing diagnosis. The results showed significant improvement in inattention and hyperactivity symptoms, as measured by psychometric indices. Further research is needed to better understand the relationship between physical activity and mental performance in adults with ADHD. Overall, the report provides a good overview of the effect of swimming on ADHD symptoms in an adult patient. However, there are some concerns that should be addressed:

Introduction:

The authors note that the prevalence of ADHD ranges from 2% to 18% and is three times higher in males than females. Provide some context or explanation for the wide range in prevalence estimates and consider adding some discussion of the potential reasons for the gender difference.

It would be useful to elaborate on how deficits in cognitive and motor functions manifest in individuals with ADHD. Providing examples or referencing relevant research would help readers better understand the challenges faced by individuals with ADHD.

Expand on the limitations and drawbacks of current treatments for ADHD, as well as the need for alternative approaches. This would provide a stronger rationale for exploring physical activity as a potential intervention and set the stage for the subsequent discussion.

The authors report that there is a limited number of studies examining the effects of physical activity in adults with ADHD, but there it would be helpful to provide some context by discussing the existing literature on physical activity interventions in children with ADHD. This could help highlight the need for further research in the adult population and emphasize the novelty of this case study.

The benefits of swimming are briefly mentioned. Strengthen your argument for using swimming as a treatment approach by providing more specific evidence or examples from previous research studies that support the positive effects of swimming on cognitive performance, mental health, motor abilities, ADHD symptoms, etc.

Case presentation:

It would be helpful to provide specific examples of the symptoms that the subject experienced to give readers a clearer understanding of her condition and its impact on her daily life.

The section provides a general overview of the swimming training program, including the frequency and duration of the sessions. However, including more specific details about the content and progression of the training program would enhance readers’ understanding of the intervention.

The scales used (BDI-II, BIS-11, and HADS-14) are briefly described. Include discussion about their psychometric properties. Are the valid measures? In addition, it would be helpful to indicate the specific time points at which the assessments were conducted to clarify the timeline of symptom changes.

Discussion:

The authors state that the results align with existing literature that indicates the positive impact of physical activity on mental health and cognitive functioning in individuals with ADHD. However, it seems necessary to clarify whether the findings can be generalized beyond this specific case. Since this study only focuses on a single subject, it is important to discuss the limitations of generalizability and the need for larger-scale studies to validate the findings.

The discussion compares the results of this case study to previous research, particularly in relation to depression, impulsiveness, and anxiety scores. While the comparisons provide useful insights, consider the inclusion of information related to the specific methodologies to help readers better understand previous research and make more informed comparisons.

The authors stated that the participant experienced a reduction in attention problems, which contradicts the literature suggesting that open motor skills are more beneficial for attention problems in ADHD. It would be interesting to explore this further and provide potential explanations.

The authors mention that most studies on the effects of swimming training on ADHD symptoms have focused on young cohorts. It would be helpful to elaborate on the potential implications of this age difference and how the findings from this adult case study contribute to the existing literature.

Other:

There are some language errors. Please copyedit carefully.

As a case report, this paper offers a unique perspective and highlights the potential benefits of swimming as a therapeutic intervention for ADHD. However, the limitations mentioned above should be addressed before consideration of publication.

There are some language errors. The authors should copyedit carefully.

Author Response

Reviewer 4

Dear reviewers

We really appreciate the time spent by the judge of our manuscript. Your instructions helped us to improve the manuscript.

C: Comment

A: Answer

C: The authors note that the prevalence of ADHD ranges from 2% to 18% and is three times higher in males than females. Provide some context or explanation for the wide range in prevalence estimates and consider adding some discussion of the potential reasons for the gender difference.

A: According to the literature there are two main factors. We provide the addition text in the manuscript (lines 32 – 35).

C: It would be useful to elaborate on how deficits in cognitive and motor functions manifest in individuals with ADHD. Providing examples or referencing relevant research would help readers better understand the challenges faced by individuals with ADHD.

A: We included some examples of the deficit in cognitive and motor function (lines 40 – 42).

C: Expand on the limitations and drawbacks of current treatments for ADHD, as well as the need for alternative approaches. This would provide a stronger rationale for exploring physical activity as a potential intervention and set the stage for the subsequent discussion.

A: Nice suggestion. Thus, we provided some extra information about the medication and physical activity treatments, enhancing the role of physical activity (lines 46 – 52).

C: The authors report that there is a limited number of studies examining the effects of physical activity in adults with ADHD, but there it would be helpful to provide some context by discussing the existing literature on physical activity interventions in children with ADHD. This could help highlight the need for further research in the adult population and emphasize the novelty of this case study.

A: We have expanded that section of the introduction, including additional references that highlight the effectiveness of physical activity in children. We have also addressed the acute effects of physical activity in adults (lines 57-63). Furthermore, we have incorporated two studies that demonstrate the beneficial effects of swimming activity in children (lines 78-85).

C: The benefits of swimming are briefly mentioned. Strengthen your argument for using swimming as a treatment approach by providing more specific evidence or examples from previous research studies that support the positive effects of swimming on cognitive performance, mental health, motor abilities, ADHD symptoms, etc.

A: Thank you for your suggestion. As we stated in the previous comment, from lines 78 to 85 we included the references of Hattabi et al. (2022) and Silva et al. (2020), who demonstrate the beneficial role of swimming activity in children with ADHD.

C: It would be helpful to provide specific examples of the symptoms that the subject experienced to give readers a clearer understanding of her condition and its impact on her daily life.

A: Nice suggestion. We discussed with the patient and she described the present example, which was included in the text (lines 95 – 101).

C: The section provides a general overview of the swimming training program, including the frequency and duration of the sessions. However, including more specific details about the content and progression of the training program would enhance readers’ understanding of the intervention.

A: Many studies that include physical activity interventions provide a brief description of the program. However, in our case, we have provided detailed information about her physical activity program from lines 114 to 134. Additionally, we have included Appendix A (Table 1), which outlines the specific drills that were used during the swimming intervention.

C: The scales used (BDI-II, BIS-11, and HADS-14) are briefly described. Include discussion about their psychometric properties. Are the valid measures? In addition, it would be helpful to indicate the specific time points at which the assessments were conducted to clarify the timeline of symptom changes.

A: We included the specific time points that the measurements were conducted in the text (lines 144 – 147) and in Table 2. Also, the reason that 2nd measurements did not conduct. Moreover, we discuss the psychometric properties and their validation (lines 148 – 177).

C: The authors state that the results align with existing literature that indicates the positive impact of physical activity on mental health and cognitive functioning in individuals with ADHD. However, it seems necessary to clarify whether the findings can be generalized beyond this specific case. Since this study only focuses on a single subject, it is important to discuss the limitations of generalizability and the need for larger-scale studies to validate the findings.

A: We understand that the results are difficult to be generalize. Therefore, we included a paragraph in the discussion (lines 251 – 256) pointing out this concern. Also, we depict it in the conclusion too (lines 270 – 276).

C: The discussion compares the results of this case study to previous research, particularly in relation to depression, impulsiveness, and anxiety scores. While the comparisons provide useful insights, consider the inclusion of information related to the specific methodologies to help readers better understand previous research and make more informed comparisons.

 A: Thank you for your suggestion. We included some methodological aspects in the studies which we compare our results to help readers better understand previous research and make more informed comparisons (lines 213 – 214; 224 – 226). 

C: The authors stated that the participant experienced a reduction in attention problems, which contradicts the literature suggesting that open motor skills are more beneficial for attention problems in ADHD. It would be interesting to explore this further and provide potential explanations.

A: This comment helped us to describe better the beneficial role of closed motor skills in ADHD patients (lines 242 – 248).

C: There are some language errors. Please copyedit carefully.

A: The errors were corrected.

Reviewer 5 Report

Very interesting case presentation! I am only curious about follow-up after the injury. How long were the benefits of the exercise sustained? And did she returned the baseline level of symptoms? I think these would be interesting to see too. However overall great work!

Author Response

Reviewer 5

Dear reviewers

We really appreciate the time spent for the judge of our manuscript. Your instructions helped us to improve the manuscript.

C: Comment

A: Answer

C: Very interesting case presentation! I am only curious about follow-up after the injury. How long were the benefits of the exercise sustained? And did she return the baseline level of symptoms? I think these would be interesting to see too.

A: Thank you for your kind words. Unfortunately, we did not have a follow-up period. However, as mentioned in the text, the patient exhibited increased ADHD symptoms (lines 136 – 140).

Round 2

Reviewer 1 Report

The manuscript has been carefully revised.

The introduction, presentation, and discussion of the results were significantly improved.

There was also a concern to draw conclusions, without neglecting the limitations of the study.

The bibliographical references were also reviewed and improved.

Thus, it is considered that the article currently meets the requirements to be published.

Reviewer 2 Report

Dear Authors,

Thank you for your revision. Now your manuscript has been improved and its impact increased.